# A Mild Dose of Aspirin Promotes Hippocampal Neurogenesis and Working Memory in Experimental Ageing Mice

**DOI:** 10.3390/brainsci13071108

**Published:** 2023-07-21

**Authors:** Jemi Feiona Vergil Andrews, Divya Bharathi Selvaraj, Akshay Kumar, Syed Aasish Roshan, Muthuswamy Anusuyadevi, Mahesh Kandasamy

**Affiliations:** 1Laboratory of Stem Cells and Neuroregeneration, Department of Animal Science, School of Life Sciences, Bharathidasan University, Tiruchirappalli 620024, India; jemiandrews1996@gmail.com (J.F.V.A.); divyabharathimsr26@gmail.com (D.B.S.); akshaykpty@gmail.com (A.K.); 2Molecular Neuro-Gerontology Laboratory, Department of Biochemistry, School of Life Sciences, Bharathidasan University, Tiruchirappalli 620024, India; sroshanaasish@gmail.com (S.A.R.); janushyas@bdu.ac.in (M.A.); 3University Grants Commission-Faculty Recharge Programme (UGC-FRP), New Delhi 110002, India

**Keywords:** aspirin, hippocampus, adult neurogenesis, doublecortin, memory, Morris water maze

## Abstract

Aspirin has been reported to prevent memory decline in the elderly population. Adult neurogenesis in the hippocampus has been recognized as an underlying basis of learning and memory. This study investigated the effect of aspirin on spatial memory in correlation with the regulation of hippocampal neurogenesis and microglia in the brains of ageing experimental mice. Results from the novel object recognition (NOR) test, Morris water maze (MWM), and cued radial arm maze (cued RAM) revealed that aspirin treatment enhances working memory in experimental mice. Further, the co-immunohistochemical assessments on the brain sections indicated an increased number of doublecortin (DCX)-positive immature neurons and bromodeoxyuridine (BrdU)/neuronal nuclei (NeuN) double-positive newly generated neurons in the hippocampi of mice in the aspirin-treated group compared to the control group. Moreover, a reduced number of ionized calcium-binding adaptor molecule (Iba)-1-positive microglial cells was evident in the hippocampus of aspirin-treated animals. Recently, enhanced activity of acetylcholinesterase (AChE) in circulation has been identified as an indicative biomarker of dementia. The biochemical assessment in the blood of aspirin-treated mice showed decreased activity of AChE in comparison with that of the control group. Results from this study revealed that aspirin facilitates hippocampal neurogenesis which might be linked to enhanced working memory.

## 1. Introduction

Aspirin is one of the most widely used generic non-steroidal anti-inflammatory drugs (NSAIDs) in the treatment regime of pain and fever [1,2]. Regular intake of aspirin provides preventive measures against various ageing-associated diseases including cardiovascular illness, cerebral stroke, thrombosis, and cancer [3]. The blockade of the cyclooxygenase (COX)-2 enzyme responsible for the synthesis of inflammatory prostaglandins is a well-recognized mode of action for aspirin [4]. While various mood disorders, neurocognitive deficits, and psychiatric illnesses have been characterized by overexpression of COX-2 in association with neuroinflammation and oxidative stress in the brain, pharmacological blockade of COX-2 has been considered a crucial neurotherapeutic intervention [5,6]. Among various COX-2 inhibitors, aspirin treatment has been speculated to enhance memory in the ageing population and mitigate neuropathogenesis in subjects with various brain diseases. While some correlative studies indicated that the association between aspirin treatment and memory is uncertain, ample scientific evidence strongly implies that aspirin prevents memory decline in elderly subjects [7,8]. Eventually, recent reports demonstrated that aspirin treatment considerably improves the synaptic plasticity in the cognitive centers of the experimental brains [9]. Moreover, the implementation of aspirin has been considered for the treatment regime of memory loss in ageing and neurodegenerative disorders including Alzheimer’s disease (AD) [10,11]. However, the underlying cellular mechanism through which aspirin modulates neuroplasticity responsible for neurocognitive measures remains unclear. Adult neurogenesis is an inimitable neuroregenerative process in which new functional neurons are continuously generated from neural stem cells (NSCs) in the hippocampus and subventricular zone (SVZ) olfactory bulb (OB) system of the brain [12,13,14]. Notably, the occurrence of neurogenesis in the hippocampus has been identified to provide the cellular basis of pattern separation, mood, spatial learning, and working memory in physiological conditions [15,16]. In contrast, memory loss upon ageing has been attributed to decreased hippocampal neurogenesis [17]. Moreover, neurodegenerative disorders have been characterized by impaired hippocampal neurogenesis due to chronic neuroinflammation accounting for various neurological deficits and dementia [12,13,18]. Therefore, targeting a centralized neuroinflammatory pathway such as COX signaling that interlinks different pathogenic mechanisms and suppresses the hippocampal neuroregenerative plasticity during ageing and neuropathogenic conditions has become an unmet therapeutic need for progressive memory loss. Considering its prominent COX inhibitory potential aspirin treatment can be presumed to be involved in the regulation of hippocampal neurogenesis in the brain that could facilitate a positive impact in boosting learning and memory. However, the scientific evidence for the effects of aspirin on the regulation of neuroregenerative potential of the brain associated with memory functions is highly limited. While aspirin has been consumed by a considerable number of individuals worldwide, understanding the neuropharmacological effects of aspirin on the modulation of neuroplasticity and neuroregenerative measures accountable for memory functions is an important scientific perusal at the preclinical state. Therefore, this study investigated the effect of aspirin on the regulation of hippocampal neurogenesis and neurocognitive behaviors in experimental ageing mice.

## 2. Materials and Methods

### 2.1. Experimental Animals

Male Bagg albino laboratory-bred (BALB)/c mice were procured from Liveon Biolabs private limited, Karnataka, India. A set of seven- to eight-months-old experimental mice (N = 12) were divided into 2 groups, namely, the control group (N = 6) and the aspirin-treated group (N = 6). Animals were maintained under standard laboratory conditions of a 12 h light/dark cycle at 22–25 °C in the animal house facility of Bharathidasan University with free access to feed and water. All animal experiments were conducted in accordance with the approval of the institutional animal ethical committee (IAEC), Bharathidasan University, under the regulations of the committee for the purpose of control and supervision of experiments on animals (CPCSEA), India (Ref No: BDU/IAEC/2017/NE/41/Dt.21.03.2017). 

### 2.2. The Treatment of Aspirin and Bromodeoxyuridine

325 milligrams (mg) of a dissolvable form of aspirin (Disprin, Reckitt Benckiser Healthcare, India) was thoroughly mixed in 500 mL of lukewarm water. Each mouse at around 32–35 g was estimated to consume around 3 mL of solution daily for 6 weeks, which is equivalent to 60 mg of aspirin/kilogram (kg) body weight (BW) per day. The dosage of aspirin was determined based on the previous report [19]. During the treatment period of aspirin from day 9 to day 13, the labelling of dividing NSCs was performed using a synthetic thymidine analog 5-bromo-2-deoxyuridine (BrdU) (Sigma-Aldrich, St. Louis, MO, USA). BrdU was dissolved in a warm sterile solution of 0.9% NaCl (Sisco Research Laboratories (SRL), Mumbai, India) and thoroughly mixed. A single dose of BrdU solution (50 mg/Kg BW) was intraperitoneally injected into all the animals in both the control and the aspirin groups for 5 consecutive days. The animals were subjected to various behavioral tests such as novel object recognition (NOR), the cued radial arm maze (cued RAM), and the Morris water maze (MWM). A camera was placed above the test fields and connected to a computer equipped with SMART 3.0 video tracking software (Pan lab, Barcelona, Spain). The entire activities of each animal were tracked, and the behavioral parameters were assessed using SMART 3.0 software. After 30 days of the last BrdU injection, experimental mice were perfused, and the brain tissues were dissected for the immunohistochemical analysis of neurogenesis (Figure 1).

### 2.3. Novel Object Recognition Test

In experimental animals, the time taken to explore a novel object has been considered a reflection of non-spatial memory function [20]. The NOR test was conducted to assess the effect of aspirin on non-spatial object memory, based on the ability of the animals to distinguish the difference between a known object and a new object in an experimental task. The experimental animals that exhibit an enhanced preference to explore a new object compared to a familiarized object have been considered to have improved memory. Initially, to assess the effect of aspirin on the recognition memory based on the ability to preference or discriminate different objects, the experimental animals were subjected to a standard NOR test as previously described by Yesudhas and colleagues (2020) [21]. During the habituation period, each animal was allowed to explore an empty arena (30 × 15 × 30 cm). Using SMART 3.0 software, the test arena was digitally divided into two equal squares, and identical cylindrically shaped, yellow-colored wooden objects, namely, A1 and A2, were placed in the middle of each square. Using SMART 3.0 software, two circular zones were digitally introduced around each object, such as zone 3 (sky blue circle) and zone 4 (pink circle). In the trial phase, each animal was released in the middle of the arena and allowed to explore and familiarize itself with identical objects for 5 min with 3 consecutive trials. After the task, the animal was returned to the home cage. After 2 h, the test phase was performed, in which object A2 was replaced by a novel, green-colored, rectangular-shaped wooden object B, but familiarized object A1 was retained at the same place. During the test phase, each mouse was released again into the arena to re-explore the objects for 5 min. The overall activities of each animal were captured using SMART 3.0. In particular, the time spent by experimental animals in the respective zones and the tendency to explore the novel objects were estimated. The discrimination index was calculated using the formula, discrimination index = (Exploration time in novel object − exploration time in familiar object)/(total exploration time in both objects) × 100 [21].

### 2.4. Morris Water Maze

The MWM is an effective behavioral test of long-term spatial learning and retention of memory in experimental rodents [22]. In the MWM task, the animals were trained to navigate from a releasing point to locate a submerged escape platform in a water pool with the use of extra maze cues or spatial signs in the behavioral room. The spatial learning performance of experimental animals was evaluated by latency to find the hidden platform upon repeated trials, and memory retention was assessed by the time spent by the animals in the target zone during the probe trial without the platform. A black circular MWM tank with a diameter of 150 cm and a depth of 50 cm was used to investigate the aspirin-mediated effect on spatial learning and memory in experimental animals, as previously described [21]. Four extra maze visual cues were placed on the walls of the behavioral room with an appropriate lighting setup. Using SMART 3.0, the whole MWM was designated as an arena, and four quadrants, namely, zone 1, zone 2, zone 3, and zone 4, were digitally introduced. For habituation, each animal was exposed to the MWM for 2 days without the platform. Then, a hidden escape platform was placed in the middle of zone 4 and submerged 1 cm beneath the surface of the water. The temperature of water in the MWM pool was maintained at around 20–22 °C. During the training session, each mouse was gently released into the water maze from four different directions in a systematic manner and trained to find the hidden platform within 1 min. At the end of each trial, the animal was allowed to stay on the platform for 30 s and then returned to the home cage. The learning session was carried out for 14 consecutive days at 4 trials for 1 min each per day with a minimum of 30 min of intervals. The tracking SMART 3.0 module was calibrated and programmed to stop automatically when the mouse reaches the platform before 1 min. For each trial, the swimming path, distance moved, speed, and time spent in each quadrant were automatically recorded during the trial and analyzed at the end of each day. The learning curve was established by the time taken to find the platform and represented as escape latency. One day after the learning session, the probe test was conducted, in which the platform was removed from zone 4, each mouse was released into the MWM pool, and the performance of the animal was recorded for 1 min. The time spent in the platform zone was considered to assess the retention memory of the animals. On the following day, the platform was placed in the opposite quadrant (zone 2) for the reversal training. Similar to the learning phase, each animal was released from all four sides systematically, and the time taken to find the submerged hidden platform in a new zone was measured. The reversal learning was carried out for 3 consecutive days consisting of 4 trials each for 1 min. The short-term learning curve for the reversal training was established to determine spatial working memory [21]. 

### 2.5. Cued Radial Arm Maze Paradigm

The use of cued RAM is highly instrumental in assessing the working memory of experimental rodents. Notably, the eight-arm radial maze with a proximal cue at the target arm enables the animals to locate flavored feed during the training session. In the test phase, the ability of experimental animals to explore the shifted proximal cue in a new location as a strategic attempt in search of feed has been considered the degree of working memory [23]. To validate the effect of aspirin on working memory, experimental animals were further challenged with cued RAM. Then, 24 h prior to the cued RAM experiment, the feed was removed from the animal cage. The maze was a standard radial maze with eight horizontal equidistantly spaced arms (44 cm length × 14 cm breadth × 12 cm height) radiating from a circular central middle zone (diameter = 32 cm). Using SMART 3.0, each arm was digitally divided into 8 segments (arm 1 to arm 8). In the learning phase, a visible wooden proximal maze cue was placed on the parapet end of arm 5, and choco flakes were placed inside arm 5. Each mouse was released in the middle of the cued RAM and allowed to explore freely all the arms. The uninterrupted movement of the mouse was recorded using SMART 3.0 for 5 min. At the end of each training, the mouse was placed in arm 5 for 30 s, and choco flakes were given as a reward. Then, the mouse was gently removed from the cued RAM and returned to the home cage. In the learning phase, 5 min × 3 trials per day were conducted for 3 consecutive days. The next day, the cue was shifted from arm 5 to arm 1 and the choco flakes were removed from arm 5. In the test phase, each animal was released in the middle and the uninterrupted moments of the animal were recorded using SMART 3.0. The time taken to enter, the number of entries, and the time spent in the newly cued arm were calculated as the measure of working explicit memory [24].

### 2.6. Animal Perfusion and Cryosection of the Brains

After the behavioral experiments paralleled by 30 days from the last injection of BrdU, the experimental mice were deeply anesthetized, and the blood samples were collected from cardiac puncture for biochemical assay, with the animals transcardially perfused with sterile 0.9% NaCl (Sisco Research Laboratories (SRL), Mumbai, India), followed by 4% paraformaldehyde (PFA) (HiMedia, Mumbai, India), as previously described [21,25]. The whole brains were dissected and soaked in 4% PFA overnight at 4 °C. The next day, brains were transferred to fresh tubes containing 30% sucrose (Sisco Research Laboratories (SRL), Mumbai, India) and stored for a week at 4 °C. Then, the brain tissues were further processed for cryosectioning using a sliding microtome (Weswox, Haryana, India) with a custom-built specimen block holder surrounded by the platform for dry ice. The Poly Freeze medium (Sigma-Aldrich, St. Louis, MO, USA) was added to the top of the specimen holder and was allowed to freeze using dry ice. Each brain was carefully separated into two hemispheres and embedded in poly freeze tissue freezing medium and kept on the specimen block holder. The brain tissue was optimally frozen using dry ice and subjected to the sagittal sections of 30 µm thickness and serially distributed in 12 tubes containing the cryoprotectant solution in a 1:1:2 ratio of glycerol (Merck, Darmstadt, Germany), ethylene glycol (Sisco Research Laboratories (SRL), Mumbai, India) in 0.1 M sodium phosphate buffer (NaH_2_PO_4_ and K_2_HPO_4_ (Sisco Research Laboratories (SRL), Mumbai, India), pH 7.5), and stored at −20 °C for further immunohistochemical analysis, as previously described [25,26].

### 2.7. Immunohistochemistry and Assessment of Neurogenesis

One out of twelve sections from the left hemisphere of the brain with 360 µm distance apart was taken free-floatingly in 12 well plates (Tarson, Kolkata, India) filled with 1× Tris-buffered saline (TBS) (Tris HCl (HiMedia, Mumbai, India) and NaCl (Sisco Research Laboratories (SRL), Mumbai, India), pH 7.4) and washed thrice using a shaker (Tarson, Kolkata, India) for 10 min each at room temperature. Next, in the antigen retrieval step, the brain sections were placed in sodium citrate buffer (10 millimolar (mM) sodium citrate dihydrate (Thermo Fisher Scientific, Waltham, MA, USA) and 0.05% Triton X (HiMedia, Mumbai, India, pH 6.0), for two hours at 65 °C in a water bath (Kemi, Kerala, India). Then, the brain sections were washed thrice in 1× TBS for 10 min each at room temperature. After washing, the sections were incubated in a blocking solution of 3% bovine serum albumin (BSA) (HiMedia, Mumbai, India) in 0.1% TBST for 1 h on a shaker at room temperature. After blocking, the primary antibodies were reconstituted in the blocking solution and added to the sections, followed by incubation for 48 h at 4 °C. To label the doublecortin (DCX)-positive immature neurons, the brain sections were incubated with rabbit anti-DCX antibody (Cell Signalling Technology, Danvers, MA, USA; 1:250 dilution) for 48 h at 4 °C. Similarly, an additional set of brain sections was processed for immunohistological labelling of microglia and incubated in a solution containing a rabbit anti-ionized calcium-binding adapter molecule (Iba)-1 antibody (Cell Signaling Technology, Danvers, MA, USA, 1:250 dilution) and incubated for 48 h at 4 °C. After 48 h, the solution containing primary antibodies was removed and washed thrice in 1× TBS for 10 min each at room temperature. Then, the brain sections were incubated with a fluorescent conjugated secondary antibody, namely, goat anti-rabbit Dylight 594 (Novus Biologicals, Littleton, CO, USA; 1:500 dilution) for 24 h at 4 °C. The next day, the solution containing secondary antibodies was removed, and the brain sections were washed twice in 1× TBS for 10 min each at room temperature. To label the cell nuclei, the brain sections were incubated with 0.1 mg/mL of 4′,6-diamidino-2-phenylindole (DAPI) (HiMedia, Mumbai, India) in TBST for 10 min. Further, the brain sections were washed in 1× TBS for 10 min and systematically arranged on the double frosted microscopic slides (Borosil, Mumbai, India) and allowed to dry overnight in the dark. Upon complete drying, the prolong glass antifade mountant (Thermo Fisher Scientific, Waltham, MA, USA) solution was applied to the brain sections on the microscopic slides and mounted with coverslips. The next day, slides were blind-coded, and the immunolabelled brain sections were visualized under a fluorescence microscope (DM750, Leica Microsystems, Wetzlar, Germany). The number of DCX- and Iba-1-positive cells was counted in the sub-granular zone (SGZ), and the granule cell layer (GCL) of the hippocampal DG regions using the ImageJ software- (version number: 1.53k) cell counter plugin and the total number of DCX-positive cells and Iba-1-positive cells per hemisphere was estimated as per previous reports [25,27]. Further, the length of the dendrites in the micrometer was measured in the DCX-positive cells in the hippocampus using ImageJ software (version number: 1.53k) with the NeuronJ plug-in and represented as an average length of the dendrites in the micrometer between control and aspirin-treated animals. The intensity of Iba-1-positive cells was measured using ImageJ software (version number: 1.53k). The captured microscopic images were used to analyze the fluorescence intensity of Iba-1-positive cells in the hippocampus between the control and the aspirin-treated groups. The measured mean intensity was represented as a percentage, as previously described [28,29].

Next, to assess the frequency of neuronal differentiation, an additional set of brain sections was taken, which were washed in 1× TBS thrice for 10 min each at room temperature. Next, the antigen retrieval step was performed in which the brain sections were placed in sodium citrate buffer for two hours at 65 °C. After antigen retrieval, the brain sections were placed in 2 normal (N) HCl solutions (Finar, Ahmedabad, India) for 10 min at 37 °C on a shaker. Then, the brain sections were treated with 0.1 molar borate buffer (boric acid (HiMedia, Mumbai, India), pH 8.5) for 10 min. Following this, the brain sections were washed thrice in 1× TBS for 10 min each at room temperature. After washing, the brain sections were incubated in 3% BSA for 1 h on a shaker at room temperature. After blocking, two primary antibodies, namely, mouse anti-BrdU (Novus Biologicals, Littleton, CO, USA; 1:100 dilution) and rabbit anti-neuronal nuclear protein (NeuN) (Novus Biologicals, Littleton, CO, USA; 1:100 dilution), were used for 48 h at 4 °C. Next, the primary antibodies were removed, and the sections were washed thrice in 1× TBS buffer for 10 min. The brain sections were incubated together with two different secondary antibodies such as sheep anti-mouse Dylight 488 (Novus Biologicals, Littleton, CO, USA; 1: 500 dilution) and goat anti-rabbit Dylight 594 (Novus Biologicals, Littleton, CO, USA) antibodies at 4 °C for 24 h. After 24 h, the secondary antibodies were removed, and the sections were washed thrice in 1× TBS for 10 min each. Finally, the brain sections were arranged on the double-frosted slides (Borosil, India) and allowed to dry overnight in the dark. After complete drying, the sections were sealed using the Prolong Glass Antifade Mountant (Thermo Fisher Scientific, Waltham, MA, USA) and dried in the dark. The brain sections were subjected to the double immunofluorescence assessment using a laser-scanning confocal microscope (LSM 710, Carl Zeiss, GMBH, Jena, Germany) at the central instrumentation facility of Bharathidasan University and the Z-stack optical images of hippocampal DG were obtained for the BrdU with NeuN. For the frequency of neuronal differentiation, 25 BrdU-positive cells (green) in hippocampal DG from each animal were assessed. The BrdU-positive cells that are co-labelled with NeuN (red) were considered double-positive cells (yellow) and multiplied by four to estimate the percentage of double-positive cells [25,30].

### 2.8. Estimation of Acetylcholinesterase Activity in the Blood

The whole blood samples from each animal were collected in a container containing an anticoagulant, 3.2% buffered trisodium citrate (Thermo Fischer Scientific, Waltham, MA, USA). The blood samples were centrifuged at 800 rpm for 10 min. The plasma was transferred into another sterile microfuge tube and then centrifuged at a speed of 2500 rpm for 15 min to obtain platelets. The resulting lower third was platelet-rich plasma (PRP) and the upper two-thirds was platelet-poor plasma (PPP). PPP was removed, and each tube was shaken to suspend the platelet pellets present at the bottom. The resulting PRP samples were subjected to the biochemical assessment of acetylcholinesterase (AChE) activity, as described by Ellman et al. in 1961 [31]. Samples were loaded in a microtiter plate and mixed with 0.1 mM acetylthiocholine iodide (Sisco Research Laboratories (SRL), Mumbai, India) and 0.5 mM 5,5′-dithiobis-2-nitrobenzoic acid (DTNB) (HiMedia, Mumbai, India). The enzyme activity of AChE in PRP was measured by the yellow color from thiocholine when it reacted with DTNB. The plate was shaken for a few seconds, and the absorbance at 412 nm was taken using a UV-VIS spectrophotometer (Synergy™ HTX Multi-Mode Microplate Reader, Biotek, Winooski, VT, USA).

### 2.9. Statistical Analyses

The values have been represented as mean ± standard error mean (SEM). Student’s *t*-test was applied to measure the statistical significance for the number of entries, duration and discrimination index for NOR test, probe test in MWM, latency and duration in cued RAM, numbers of DCX- and Iba-1-positive cells, length of the dendrites, the intensity of Iba-1-positive cells, percentage of BrdU/NeuN double-positive cells, and AChE activity between the control and aspirin-treated groups. The escape latency to find the platform and reversal learning in the MWM test, the learning curve in cued RAM, were assessed by one-way analysis of variance (ANOVA) followed by Tukey’s post hoc test for multiple comparisons. All the statistical analyses were made using Graph Pad Prism. The significance level was assumed at *p* < 0.05 unless otherwise indicated. In the data, the statistical significance * indicates *p*-value < 0.05, ** indicates *p*-value < 0.01, and *** indicates *p*-value < 0.001, along the degree of freedom (df), and F value (F).

## 3. Results

### 3.1. Aspirin Treatment Improved Novel Object Recognition in Experimental Mice

During the training session, experimental mice were exposed to a test arena and freely allowed to explore two identical objects. In the test phase, experimental mice were exposed to the replacement of a familiar object with a novel object (Figure 2A). Though mice in both the control and the aspirin-treated group were able to recognize the novel object, experimental mice that were treated with aspirin showed an enhanced tendency to visit the novel object zone and explored the new object more than the familiarized object compared to the mice in the control group (control = 3.5 ± 0.67 vs. aspirin = 6.5 ± 0.34, visits, ** *p*-value = 0.0026, df = 10, F = 3.857) (Figure 2B). As a result, aspirin-treated mice spent significantly more time in the novel object zone than the control group (control = 35 ± 4.4 vs. aspirin = 54 ± 5.1 s, * *p*-value = 0.0157, df = 10, F = 1.360) (Figure 2C). Eventually, the percentage of the discrimination index was significantly enhanced in the aspirin-treated group compared to the control group (control = 24.2 ± 3.6 vs. aspirin = 38 ± 3.6, * *p*-value = 0.0229, df = 10. F = 1.031) (Figure 2D), thereby indicating that aspirin treatment improved recognition memory in experimental ageing mice (Figure 2).

### 3.2. Aspirin Treatment Enhanced Spatial Learning and Working Memory in the Morris Water Maze Task

In the MWM task, the preference of experimental mice to find the hidden platform was gradually improved during the 14 days of training period in both control and aspirin-treated groups (Figure 3A). Assessment of the data obtained from the learning curve indicated that the time taken to find the platform or escape latency was decreased in the aspirin-treated group compared to the control group, noticeably from the fifth day of training (Figure 3D). Next, the experimental animals were challenged for the potential of memory retention during the probe test. The platform was removed from the MWM, and each animal was released to the pool; the time spent in the platform zone was measured for 1 min. The average duration of search to find the platform in the target quadrant by the aspirin-treated animals was found to be significantly increased in comparison with that of control animals (control = 14 ± 1.8 vs. aspirin = 27 ± 3.4 s, ** *p*-value = 0.0095, df = 10, F = 3.715) (Figure 3B,E). Next, to assess the reversal learning as a measure of working memory, the platform was placed in the opposite quadrant, and the experiment was conducted for the next three days. Eventually, the time taken to find the platform in the opposite quadrant was reduced by experimental mice in the aspirin-treated group than the control group (day 1: control = 51 ± 2.5 vs. aspirin = 37 ± 2.2 s, ** *p*-value< 0.01, df = 46, F = 1.313; day 2: control = 37 ± 3.6 vs. aspirin = 23 ± 1.9 s ** *p*-value < 0.01, df = 46, F = 2.013; day 3: control = 28 ± 3.0 vs. aspirin = 12 ± 1.7 s, *** *p*-value < 0.001, df = 46, F = 3.000) (Figure 3C,F). Taken together, the MWM test indicates that aspirin treatment improves spatial learning and working memory in experimental mice (Figure 3).

### 3.3. Aspirin Treatment Improved Working Memory in Cued RAM

In the trial phase of RAM, the time spent by the experimental mice in both the control and aspirin-treated groups in the cued arms were nearly similar at the beginning of the experiment, but the animals in the treatment group showed slight improvement upon training (Day 1: control = 250 ± 20 vs. aspirin = 230 ± 7.8 s, *p*-value = 0.3528, df = 46, F = 6.709; Day 2: control = 131 ± 19 vs. aspirin = 101 ± 22 s, *p*-value = 0.3048, df = 46, F = 1.381; Day 3: control = 111 ± 18 vs. aspirin = 89 ± 8 s, *p*-value = 0.2752, df = 46, F = 4.973) (Figure 4A,C). However, during the test phase, while the latency to enter the cue-shifted arm was reduced (control = 207 ± 29 vs. aspirin = 131 ± 15 s, * *p*-value = 0.0396, df = 10, F = 3.670) (Figure 4B,D), time spent in the newly cued arm was significantly increased in the aspirin-treated group in comparison with that of the control (control = 22 ± 4 vs. aspirin = 38 ± 5 s, * *p*-value = 0.0417, df = 10, F = 1.617) (Figure 4B,E). Taken together, the cued RAM experiment validates that the aspirin treatment facilitated working and declarative memory in experimental mice (Figure 4).

### 3.4. Aspirin Treatment Increased the Hippocampal Neurogenesis

In the stem cell niches of the mammalian brain, the number of DCX-positive immature neurons has been considered to reflect ongoing neurogenesis [32]. The immuno labelling study of the DCX-positive cells in the hippocampal DG revealed that the total number of immature neurons was significantly increased in the aspirin-treated group in comparison with that of the control group (control = 3742 ± 420 vs. aspirin = 6087 ± 537, number of cells, ** *p*-value = 0.0064, df = 10, F = 1.635) (Figure 5A,C). Further, in the morphological assessment, the length of the dendrites of DCX-positive cells was found to be increased in the hippocampal DG region of the brains of the experimental mice in the aspirin-treated group in comparison with that of the control group (control = 90 ± 9.3 vs. aspirin = 172 ± 17.2, µm, ** *p*-value = 0.0019, df = 10, F = 3.392) (Figure 5B,D). BrdU labelling of newly dividing NSCs followed by the co-expression with a mature neural mark has been a critical step in assessing the neuronal fate in the hippocampus of the brain [25]. In the confocal microscope-based co-immunolabelling study, the percentage of BrdU/NeuN double-positive cells was significantly increased in the granule cell layer of hippocampal DG regions of the brains of the experimental mice in the aspirin-treated group in comparison with that of the control group (control = 76 ± 3.3 vs. aspirin = 91 ± 1.3, % BrdU-/NeuN-positive cells, ** *p*-value = 0.0020, df = 10, F = 6.088) (Figure 5E,F). Taken together, aspirin treatment appeared to promote neuronal differentiation of NCS, improve dendritic length and arborization, and facilitate hippocampal neurogenesis in experimental mice (Figure 5).

### 3.5. Aspirin Treatment Decreased the Abnormal Number of Microglial Cells in the Hippocampus and Reduced the Activity of AChE in the Blood of the Experimental Animals

Iba1 is a prominent marker of microglial cells in the brain [26]. Ageing related to memory decline has been reported to be associated with an abnormal number of microglia responsible for neuroinflammation [33]. Notably, in the qualitative and morphological observation of the Iba-positive cells in the hippocampus, the activated signs of microglia such as dystrophic ameboid shapes seen in hippocampal DG ageing animals were found to be less pronounced in the aspirin treatment. Moreover, the quantification of the Iba-positive cells indicated a significant reduction in the number of microglia in the hippocampal DG of mice in the aspirin-treated group compared to the control (control = 844 ± 26 vs. aspirin = 612 ± 39, number of Iba1-positive cells, ** *p*-value = 0.0018, df = 10, F = 2.505) (Figure 6A,B). Also, the mean intensity of Iba1 immunoreactivity was significantly decreased in the hippocampus of mice in the aspirin-treated group than in the control group (control = 100 ± 10 vs. aspirin = 55 ± 8, percentage of mean intensity ** *p*-value = 0.0029, df = 10, F = 5.336) (Figure 6C). Recently, the enhanced activity of AChE in the blood has been considered to negatively correlate with the degree of memory function [34]. The colorimetric determination of AChE activity was performed to measure the increase in the intensity of the yellow color produced from thiocholine when it reacts with dithiobisnitrobenzoate ions. The results revealed that the absorbance values of AChE activity in blood samples of aspirin-treated animals were significantly reduced when compared to that of control animals (control = 1.8 ± 0.07 vs. aspirin = 1.3 ± 0.08, the absorbance of AChE activity, ** *p*-value = 0.0010, df = 10, F = 1.038) (Figure 6D).

## 4. Discussion

Progressive memory loss is one of the increasing clinical concerns worldwide, posing major challenges to the ageing population, healthcare providers, and biomedical research sectors [13,35,36]. While investigations into the underlying mechanisms and therapeutic targets have been the ongoing preclinical quest, the identification of drugs that exert precognitive action has become an unmet need [13,37]. Aspirin is a widely used painkiller that blocks the activity of COX-2, thereby mitigating inflammatory processes in various diseases [38,39]. Moreover, aspirin has also been identified to modulate the biochemical pathways related to cholesterol rafts, proton pumps, capsaicin receptors, N-methyl-D-aspartate (NMDA) receptors, voltage-gated calcium channels, and peroxisome proliferator-activated receptor alpha (PPAR-α) [40,41,42,43,44,45]. For the past few decades, the beneficial effects of aspirin treatment against cerebrovascular diseases have clearly been established [46,47,48]. Numerous experimental studies suggest that aspirin treatment contributes to brain functions as it positively modulates behavioral, biochemical, and cellular parameters [49]. Based on the animal behavioral and immunohistochemical assessments, the present study demonstrates that aspirin treatment improves working memory in correlation with the increased level of neurogenic process and reduced number of microglial cells in the hippocampal DG of the experimental ageing mice brains.

The available data on the effect of aspirin on the behavioural outcome of cognitive functions appear to be inconsistent. While few data indicate that aspirin treatment is ineffective to modulate memory in certain situations, ample scientific evidence strongly advocates that aspirin treatment effectively modules neurocognitive behaviors. Few randomized placebo-controlled trials could not establish the positive effect of aspirin on cognitive decline [50,51]. A study by A Chang et al highlighted that aspirin is not effective against Paclitaxel, an anticancer drug induced memory loss in experimental mice [52]. However, a water maze-based study by Smith JW and colleagues (2002), reported that a modest 6 weeks treatment of aspirin improves the cognitive functions in aged experimental rats [53]. A similar study by Saiman Rizvan et al 2016 revealed that a mild dose of aspirin can improve spatial memory in association with the alteration of δ-opioid receptor expression in the cortex of the mouse model of AlCl_3_- induced neurotoxicity [54]. A very recent study indicated that a mild dose of aspirin treatment improves recall long-term memory in lipopolysaccharide (LPS) induced inflammation in an experimental model of pond snail [55]. Considering its neuroprotective nature, a chitosan-encapsulated drug delivery-based study strongly supports the memory-enhancing capacity of aspirin in an APP/PS1 transgenic mice model of AD [56]. The differential effect of aspirin-mediated change in cognitive behaviours has been speculated to be dose-dependent. However, the report on the effect of aspirin hippocampal plasticity responsible for learning and memory is limited. Based on the data derived from NOR, MWM, and Cued RAM, and the immunohistochemical-based study of neurogenesis, the present study strongly supports and provides a new foundation for the memory-enhancing capacity of low-dose aspirin due to its pro-neurogenic and antiinflammatory capacity in the hippocampus of experimental mice. 

In response to ageing-mediated biological changes and neurodegenerative events, proinflammatory molecules are discharged from activated immune cells such as microglia in the brain [12,25]. The abnormally elevated levels of proinflammatory molecules that are discharged from an abnormally increased number and activated microglia have been known to impair the neurogenic process at the level of proliferation, differentiation, and survival of NSCs, leading to progressive memory impairments [12,57]. Notably, aspirin has been reported to induce the production of endogenous lipoxins, a potent anti-inflammatory molecule, which diminishes the pathogenic activation of microglia, thereby reducing the load of various inflammatory molecules, including C-reactive protein tumor necrosis factor (TNF)-α and interleukin (IL)-6 in experimental animals [58,59,60]. Elevated levels of transforming growth factor (TGF)-beta, a key component of neuroinflammation in the brains of ageing subjects and neurodegenerative conditions, have been linked to impaired neurogenesis in the hippocampus at the level of NSC proliferation, resulting in dementia [12,25]. Therefore, the blockade of TGF-beta secretion and its downstream signaling pathway has been proposed as a therapeutic approach to activate the neurogenic process in the brain to alleviate dementia [12]. Notably, aspirin has been reported to suppress the secretion and signaling of TGF-beta in various organs during pathogenic conditions [61,62]. Considering the facts, aspirin-mediated enhancement in the hippocampal neurogenesis might be related to its interference with enhanced TGF-beta secretion and Smad signaling resulting from ageing-related changes. Therefore, this study supports the fact that aspirin reduces neuroinflammation through the inactivation of microglia, which might facilitate the neuronal differentiation of NSCs followed by their efficient integration in the hippocampus. Therefore, the immunomodulatory function of aspirin in the brain could be a primary extrinsic reason for the improving regenerative plasticity responsible for enhanced memory in experimental and human subjects. Recently, an in vitro study by Giacomo Pozzoli et al. reported that aspirin promotes the differentiation of SK-N-SH (N) human neuroblastoma cells through the modulation of the expression of cell cycle checkpoint markers p21^Waf1^ and Rb1, independent of the COX pathway [63]. Moreover, ample experimental evidence supports that low-dose aspirin induces cell cycle arrest and induces differentiation and growth in many cells [63,64,65]. Therefore, the direct action of aspirin on the cell cycle regulation of NSC may not be excluded. In this study, the immunohistochemical assessments of DCX-positive cells and confocal-based BrdU/NeuN revealed a concomitant increase in the neurogenic process at the level of neuronal differentiation of NSCs in the hippocampus of ageing experimental mice. Notably, dendritic arborization of DCX-positive cells was found to be increased in the hippocampal DG of aspirin-treated animals. The augmented dendritic arborization and length signifies the state of neural differentiation and integration of DCX-positive cells in the hippocampal circuit [66,67]. Moreover, a hippocampal organotypic culture-based study revealed that aspirin treatment contributes to synaptic formation by enhancing the spine density of neurons [68]. Considering the facts, it can be hypothesized that the procognitive effect of aspirin might collectively result from the enhanced neuronal differentiation of NSCs and augmentation of dendritic and synaptic strength in the newly formed neurons in the hippocampus of the brain. However, few randomized controlled trials and meta-analyses have stated no association between aspirin and cognitive functions [7,69]. Thus, there exists an ongoing considerable debate about the beneficial effects of aspirin against dementia and neurocognitive impairments [70]. Nevertheless, convincing experimental evidence supports the procognitive effect of aspirin as it appears to positively modulate hippocampal plasticity. In the experimental models challenged with glutamate excitotoxicity and hypoxia, aspirin treatment has been reported to provide neuroprotection in the brain [71]. Aspirin treatment has been reported to mitigate free radical production, thereby mimicking the antioxidant potential in different organs including the brain [49,72]. In addition, aspirin treatment appears to be associated with the epigenetic modification upregulation of the expression of BDNF that is responsible for preserving the memory function in benzo[a]pyrene (BaP)-treated experimental mice [73]. 

Notably, the neuroprotective role of aspirin has been linked to the inhibition of NF-kB signaling in the brain [74]. Depeng Feng et al. reported that aspirin facilitates neuroprotection in experimental animal models treated with lipopolysaccharide (LPS), as well as in cerebral ischemia–reperfusion (CIRP) injury [75]. Patel et al. demonstrated that aspirin interacts with PPARα and concedes memory functions through cAMP response-element-binding protein (CREB)-mediated hippocampal plasticity [9,45]. Eventually, cohort- and population-based studies revealed an inverse relationship between long-term intake of low-dose aspirin and the severity of dementia [69,76]. Chandra et al. (2018) reported that a low dose of aspirin decreases amyloid load through the activation of peroxisome-proliferator-activated receptor alpha (PPAR-α) in the hippocampus of a 5× FAD mouse model of memory loss [77]. Saima Rizwan et al. reported that a low-dose aspirin treatment improved memory in association with an alteration of the opioid system in an experimental mouse model of memory loss [54]. While the circulating platelets have been recognized to contribute to amyloid pathology in the brain upon ageing, the platelet-reducing effect of aspirin has been considered to minimize the neuropathogenic events in the brain of subjects with neurodegenerative disorders [70,78,79]. 

Notably, regulation of neurogenesis in the hippocampus has been linked to levels of various neurotransmitters that are crucial for learning and memory. Acetylcholine (ACh) is one of the key neurotransmitters that play a key role in learning and memory [21,80]. Notably, many neurodegenerative disorders have been characterized by abnormal levels of ACh in the circulation and the brain [81]. As age increases, a decline in the level of ACh is evident in the circulation and the brain [21,82]. Remarkably, increased activity of AChE, an enzyme that hydrolyses ACh, has been linked to the pathomechanism of cognitive impairments and memory loss in various neurocognitive diseases, including AD [83,84]. Eventually, increased levels of AChE in the blood have been considered a biomarker of dementia [34,85]. Thus, the implementation of AChE inhibitors has been considered to mitigate memory loss in ageing and AD [86]. Recently, few COX inhibitors have been known to diminish the activities of AChE [87]. Notably, bioinformatics-based docking studies revealed that aspirin binds to the active sites of AChE and acts as a potential inhibitor of its activities [88,89]. Ample reports suggest that inhibition of AChE facilitates hippocampal neuroregenerative plasticity in the adult brain [90]. Considering the aforementioned factors, it can be hypothesized that aspirin treatment might play a role in the regulation of hippocampal regenerative plasticity responsible for memory enhancement in association with the inactivation of AChE. Therefore, this study supports and provides insight into the procognitive, proneurogenic effects of aspirin that can be considered to translate for the treatment regimens to boost neuroregenerative plasticity noticed during various disease conditions with neurocognitive impairments.

## 5. Conclusions

The present study demonstrates that aspirin treatment enhances working memory in ageing experimental mice in association with enhanced neuronal differentiation in the hippocampus. The present study provided evidence that aspirin treatment is detrimental to abnormal microglia in the hippocampus, which could in part play a crucial role in the positive regulation of the neurogenic process in the brain. The memory-enhancing property of aspirin might be associated with the positive regulation of neuroregenerative plasticity in the hippocampus. This study reveals that aspirin has the potential to inhibit the activity of AChE in circulation, a pathogenic molecular marker that has been known to be associated with dementia. Thus, aspirin-mediated restoration of cholinergic functions responsible for cognitive function could be highly relevant. The present study denotes the necessity for further experiments to reveal the mechanisms responsible for the pro-neurogenic effects of aspirin in the brain, as they could reveal the therapeutic significance to preventing memory decline and treating dementia. Notably, some reports indicate that aspirin therapy is less effective in women than in men [91]. The reason for the differential effect of aspirin between males and females remains unclear. However, future studies are required to validate the effect of aspirin on neurodegenerative plasticity in female animals.

## Figures and Tables

**Figure 1 brainsci-13-01108-f001:**
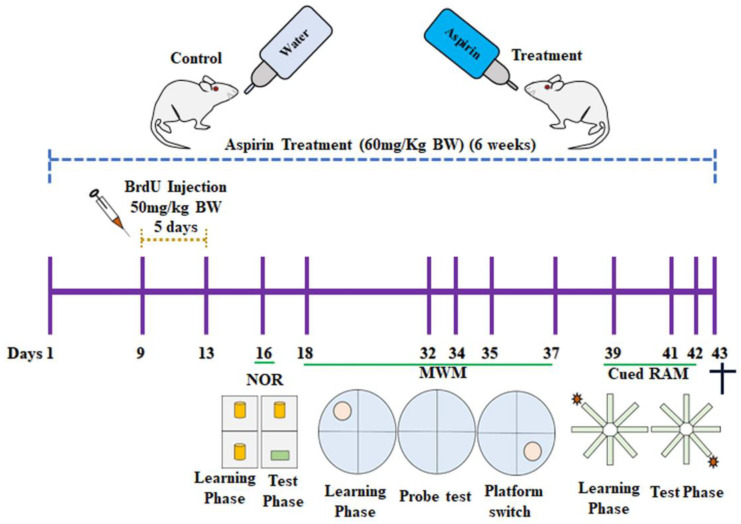
Illustration representing the experimental design of the study. The † symbol represents that the animal has been sacrificed.

**Figure 2 brainsci-13-01108-f002:**
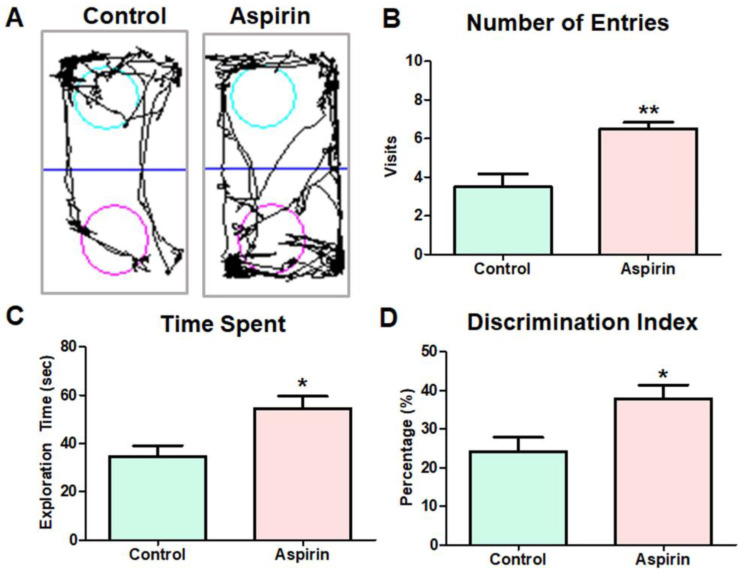
Novel object recognition (NOR) test. (**A**) Representative tracking image of the NOR test of control and aspirin-treated mice. The blue line indicates the digital separation of zone 1 and zone 2 using the SMART 3.0 video tracking system. The sky-blue-colored circle indicates the familiar object and the pink-colored circle indicates the novel object. (**B**) The bar graph represents the number of entries into the novel object zone (** *p*-value = 0.0026, df = 10, F = 3.857). (**C**) The bar graph represents the time spent in the novel object zone (* *p*-value = 0.0157, df = 10, F = 1.360). (**D**) The bar graph depicts the percentage of the discrimination index (* *p*-value = 0.0229, df = 10. F = 1.031).

**Figure 3 brainsci-13-01108-f003:**
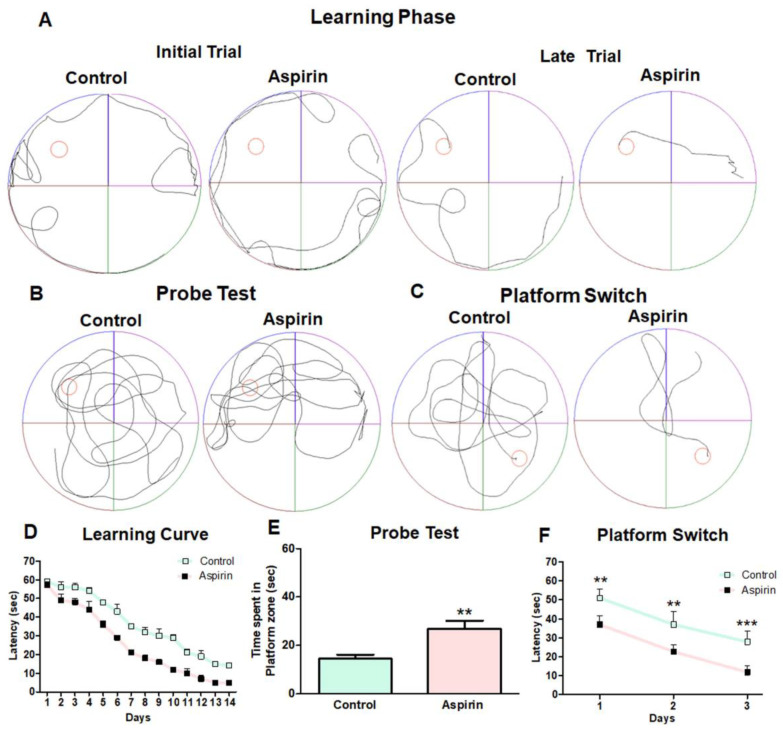
Spatial learning and memory of animals in the Morris water maze (MWM) test. (**A**) Representative tracking image of MWM during the initial and late stages of the learning phase. The purple colour indicates zone 1, the green colour indicates zone 2, the brown colour indicates zone 3 and the blue colour indicates zone 4 and the red colour indicates the platform. (**B**) Representative tracking image of MWM during the probe test. (**C**) Representative tracking image of MWM during platform switch. (**D**) Result of the learning curve with escape latency to find the hidden platform. (**E**) The bar graph describes the time spent by the animals in the platform zone during the probe test (** *p*-value = 0.0095, df = 10, F = 3.715). (**F**) The learning curve with escape latency to find the hidden platform in the opposite quadrant during the platform switch (Day 1: ** *p*-value < 0.01, df = 46, F = 1.313; Day 2: ** *p*-value < 0.01, df = 46, F = 2.013 Day 3: *** *p*-value< 0.001, df = 46, F = 3.000).

**Figure 4 brainsci-13-01108-f004:**
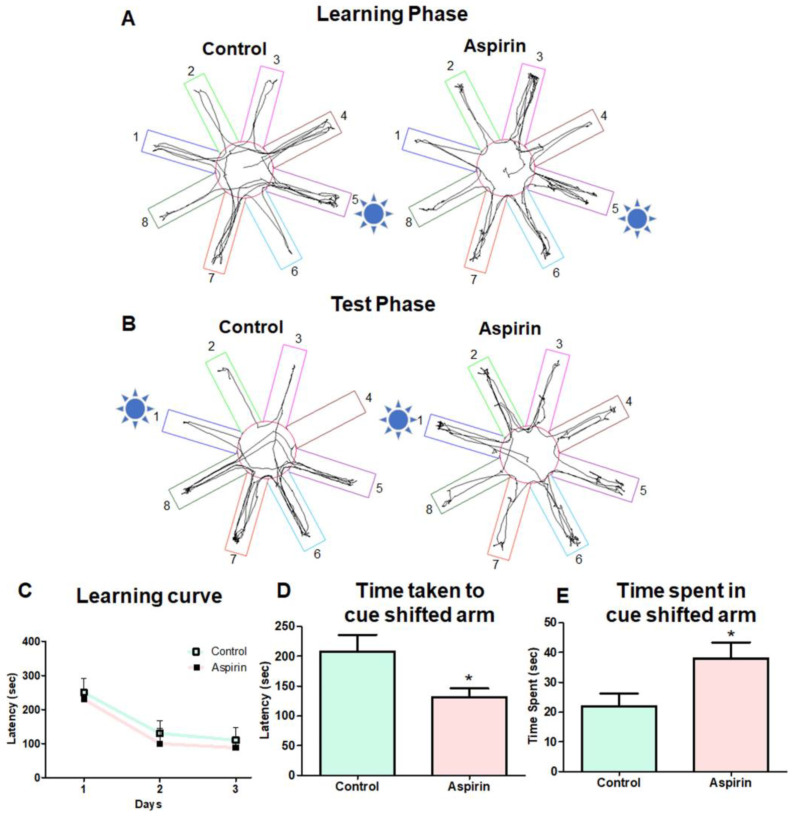
Cued radial arm maze test. (**A**) Representative tracking image of cued RAM at the learning phase. The numbers 1 to 8 in the figure denotes order of the arms in the RAM. The sun symbol represents the proximal cue in arm 5 in which rewards were placed in the learning phase. (**B**) Representative tracking image of cued RAM at the test phase, where the sun symbol represents the proximal cue in which the cue was replaced in arm 1 and the food was removed in the test phase. (**C**) Result of the learning curve with latency to reach the arm in which cue and food were placed (Day 1: *p*-value = 0.3528, df = 46, F = 6.709) Day 2: *p*-value = 0.3048, df = 46, F = 1.381); Day 3: *p*-value = 0.2752, df = 46, F = 4.973). (**D**) The bar graph represents the time taken to reach the cue-shifted arm by the animals in the test phase (* *p*-value = 0.0396, df = 10, F = 3.670 ). (**E**) The bar graph describes the time spent in the cue-shifted arm by the animals in the test phase (* *p*-value = 0.0417, df = 10, F = 1.617).

**Figure 5 brainsci-13-01108-f005:**
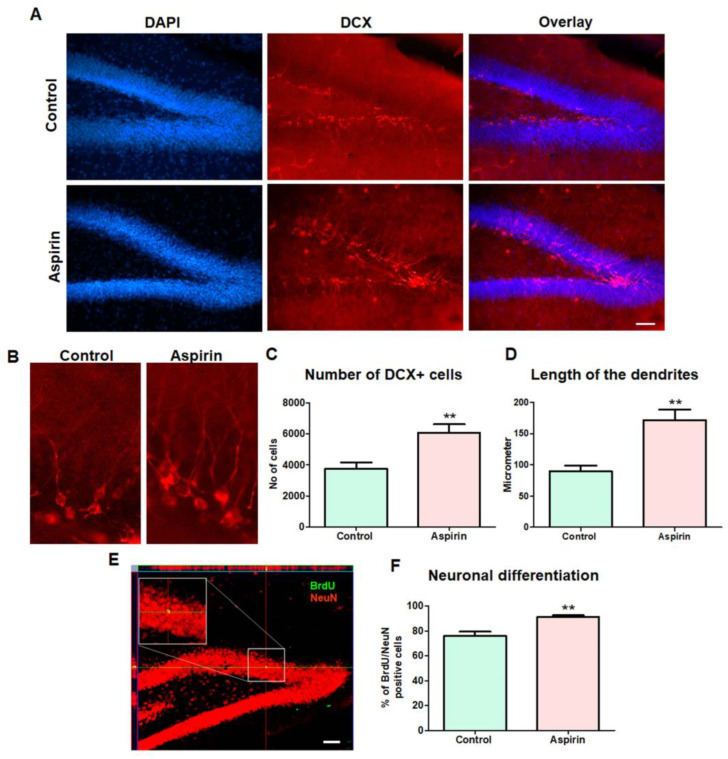
Immunohistochemical assessment of immature neurons. (**A**) Representative image of DCX-positive cells in the dentate gyrus (DG) of the experimental animals. The image illustrates the DAPI, DCX-positive cells, and overlay in the DG of the hippocampus. The scale bar = 25 µm. (**B**) The enlarged image illustrates the length of the dendrites. (**C**) The bar graph represents the number of DCX-positive cells in the dentate gyrus (** *p*-value = 0.0064, df = 10, F = 1.635) (DG) of the experimental animals. (**D**) The bar graph describes the length of the dendrites in the dentate gyrus (DG) of the experimental animals (** *p*-value = 0.0019, df = 10, F = 3.392). (**E**) Representative confocal image of BrdU/NeuN double-positive cells in the DG of the hippocampus. (**F**) The bar graph represents the percentage of BrdU/NeuN double-positive cells (** *p*-value = 0.0020, df = 10, F = 6.088). The scale bar = 25 µm.

**Figure 6 brainsci-13-01108-f006:**
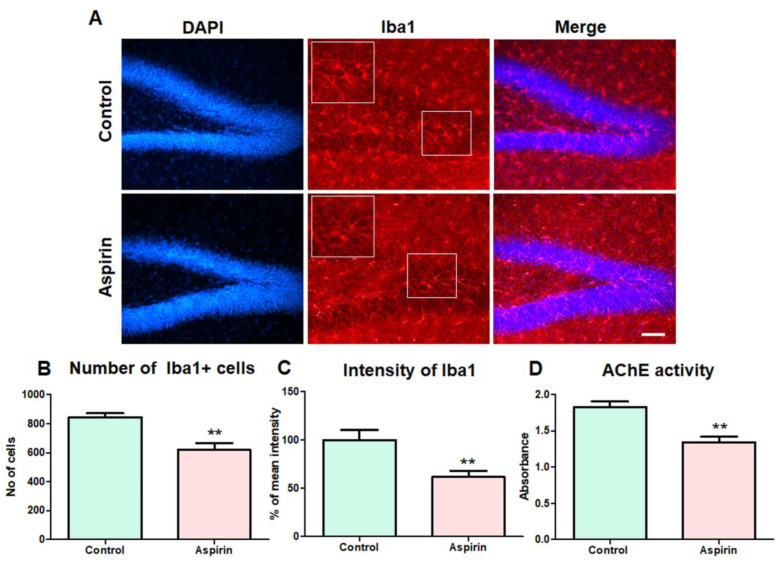
Immunohistochemical assessment of microglia in the hippocampus and assessment of AChE activity in the blood. (**A**) Representative image of Iba1-positive cells in the dentate gyrus (DG) of the experimental animals. The image represents DAPI, Iba1-positive cells, and overlay in the DG of the hippocampus. The scale bar = 25 µm. (**B**) The bar graph represents the number of Iba1-positive cells in the dentate gyrus (** *p*-value = 0.0018, df = 10, F = 2.505) (DG) of the experimental animals. (**C**) The bar graph depicts the percentage of mean intensity of Iba1-positive cells in the dentate gyrus (** *p*-value = 0.0029, df = 10, F = 5.336). (**D**) The bar graph represents the enzymatic activity of acetylcholine esterase in the blood samples of the experimental animals (** *p*-value = 0.0010, df = 10, F = 1.038).

## Data Availability

All data needed to evaluate the conclusions are present in the paper.

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
