# Peer review of "A Mild Dose of Aspirin Promotes Hippocampal Neurogenesis and Working Memory in Experimental Ageing Mice"

_brainsci, 2023, doi:10.3390/brainsci13071108_

Round 1
Reviewer 1 Report
This study assessed the effects that aspirin has on aging experimental mice through both behavior and biochemical means. The research shows that aspirin given to 7-8 month old male mice had positive effects in both behavior (spatial and working memory) as well as in markers of inflammation.
2.1: Was a power analysis completed for this study? An N of 12 with 2 groups of 6 animals each seems fairly low for a behavioral experiment. How did the researchers determine that the total sample size of 12 was needed and why only male mice?
2.2: The authors state that each mouse in the treatment group was given a daily dose of aspirin through the drinking water – were the animals group housed? Individually housed? How was intake monitored? Was water weighed or consumption recorded to verify how much aspirin was actually received? Also, how long was aspirin given? In line 92 the authors say that “during the treatment period of aspirin from day 9 to day 13”… are we to then assume that the aspirin was given for 13 days? Or for 4 days (9-13)? This needs to be clarified (e.g. exactly how many days were mice given aspirin).
2.3: For the probe trial, what starting position was the animal released in? A previous starting position or a novel starting position? Also, what’s the rationale for a 14 day acquisition period? Numerous MWM paradigms use much shorter training times (just a quick search in literature reveals numerous examples of mice in the MWM with reduced trials, including Barnhart et al. (2015), Dimopoulous et al. (2022), Rodriguez & Lippi (2022), Vorhees & Williams (2006); typically not a lot seen that ever go beyond 8 days of training).
2.4: What is the time between behavioral tasks? For instance, how many days after the MWM did you wait before doing the cued RAM?
Line 193: should be micrometer, not micromolar (The capital M denotes Molarity, not meter as in distance)
Line 269: AChE leads to the breakdown of acetylcholine into acetic acid and choline; can the authors please modify this or expand on what they currently have in place? Authors also state in line 518 that Ach breaks down into choline and acetate – so please rework these sections
3.1
Lines 287-295 could all be moved and incorporated into the methods section; the results should start discussing what was actually found
Rephrasing of your results to make it more clear: (e.g. Lines 299-300: As a result, aspirin-treated mice spent significantly more time in the novel object zone than the control group). Also, please add the p values for these results and appropriate t statistics as well (t value, degrees of freedom, etc.) You also state the units in the figure, but you should also state that these are seconds when you’re reporting the means (and I’m assuming you’re also reporting standard deviations? Or are those SEMs? Please clarify).
3.2
Lines 313-319 could all be moved and incorporated into the methods section; the results should start discussing what was actually found
P values for all given analyses (latency, probe, reversal) are needed in your results; additionally, all F statistics should be provided as well (degrees of freedom, F value, etc.)
For figure 2, In figure captions, the authors should denote what the astericks (**, ***) mean with a legend, **p < 0.01, ***p < 0.001 (for instance).
Line 335: You state aspirin mice took 112 +/- 8; did you mean 12 or 11? This is a very large jump
3.3 (Line 345 – it should be named 3.3 since this is following MWM which is 3.2)
The researchers absolutely need to provide p values as well as t test results in reporting of results.
For figure 3C, if there is no data to present for day 4, the authors should remove 4 from the x axis.
In figure captions (this and the others), the authors should denote what the astericks (*) mean with a legend, *p < 0.05 (for instance).
The authors should also be sure to put in the units of measurement in all reported results (for instance, 250 +/- 99 – what are the units being displayed here)?
In results, the specific figure (for example, Fig 5a, Fib 5b) should be stated after discussion of that specific result, instead of just saying (Fig 5) at the very end of the result paragraph
Line 547: “…reveal the mechanisms responsible for the …”
Overall comment: authors need to remark on 1) why female mice were not also included, and 2) has there been any research investigating aspirin given to female experimental mice or female human subjects?
There were areas of minor concern (grammar) that authors should review before final approval/publication.
Author Response
Reviewer 1
- This study assessed the effects that aspirin has on aging experimental mice through both behavior and biochemical means. The research shows that aspirin given to 7-8-month-old male mice had positive effects in both behaviors (spatial and working memory) as well as in markers of inflammation.
Response: We express our sincere thanks to reviewer 1 for constructive remarks and suggestions which are highly valuable and helpful for the substantial improvement of the manuscript. We have addressed all the queries of reviewer 1 and carefully revised the overall content and improved the accuracy of the manuscript as per the suggestions.
1.1: Was a power analysis completed for this study? An N of 12 with 2 groups of 6 animals each seems fairly low for a behavioral experiment. How did the researchers determine that the total sample size of 12 was needed and why only male mice?
Response: We thank reviewer 1 for the valid question. The number of animals used per experimental group has been designated by the Institutional Animal Ethic Committee (IAEC) under the stringent guideline of experts from the Committee for the Purpose of Control and Supervision of Experiments on Animals (CPCSEA), India. We admit that N=6 per experimental group is moderate for the behavioural study. Therefore, substantial number of trials were assigned to each animal in the behavioural experiments. The protocol has been well established for the adequate set of animals and reported in our previous publications (Selvaraj et al 2023 and Yesudhas et al 2020).
Due to oestrus cycle in female mice, different animals in the same experimental group might display different oestrus phases and vary with the hormonal status which could differentially influence adult neurogenesis and neurocognitive function. Hence this study has been restricted and devoted to assess the effect of aspirin in male animals. However, future studies are required to validate the effect of aspirin on neurodegenerative plasticity in female animals too as few reports indicate that aspirin therapy is less effective in women than in men. We incorporated a segment of text to cover this notion in the discussion part.
1.2: The authors state that each mouse in the treatment group was given a daily dose of aspirin through the drinking water –were the animals group housed? Individually housed? How was intake monitored? Was water weighed or consumption recorded to verify how much aspirin was actually received? Also, how long was aspirin given? In line 92 the authors say that “during the treatment period of aspirin from day 9 to day 13” are we to then assume that the aspirin was given for 13 days? Or for 4 days (9-13)? This needs to be clarified (e.g. exactly how many days were mice given aspirin).
Response: We thank reviewer 1 for the insightful comments and remarks. We have housed the mouse in groups. In one cage, three mice were grouped and maintained. The consumption of aspirin-containing water by all animals was monitored daily by an experimenter and assessed by measurement of the volume of water in the feeding bottle every 24 hours. The approximate measurement of water consumed by a mouse was calculated daily (around 3 mL/animal/day) which containing 60 mg of aspirin /kg BW. This paradigm was adopted from the previous studies by Sandy Chan et al., PLoS One. 2019 (https://doi.org/10.1371/journal.pone.0204295). With reference to the duration of treatment, aspirin was given orally in the drinking water from day 1 till sacrifice. In addition, the aspirin-consuming animals were intraperitoneally injected with BrdU for 5 consecutive days (Between day 9- to day 13) to assess the frequency of neuronal differentiation accounting for newly born neurons in the hippocampus. We have added an extra figure to clearly depict the experimental design in the revised manuscript.
1.3: For the probe trial, what starting position was the animal released in? A previous starting position or a novel starting position? Also, what’s the rationale for a 14-day acquisition period? Numerous MWM paradigms use much shorter training times (just a quick search in literature reveals numerous examples of mice in the MWM with reduced trials, including Barnhart et al. (2015), Dimopoulous et al. (2022), Rodriguez & Lippi (2022), Vorhees & Williams (2006); typically, not a lot seen that ever go beyond 8 days of training).
Response: We thank and agree to reviewer 1 on the clarifications. Morris water maze is generally used for the assessment of the long-term acquisition of spatial learning and memory. However, there are both intensive short and durable long training period in MWM. Both the short-term and long-term paradigms have been well established, accepted, and practiced but the protocol of learning period varies among different laboratories. In a shorter period of learning (about one week), the number of sessions would be more per day which might be physically exhaustive for the animals. Eventually, some animals in the group may require more days to learn. Whereas in the long period of learning (two weeks), the number of trials (4 trials/day) would be optimum per day and all animals will get enough stretch of time to learn in finding the hidden platform. Moreover, the focus of the study is to assess the effect of aspirin on neurogenesis associated with memory in which the BrdU paradigm to assess adult neurogenesis requires a longer time. To cover this, we have chosen 14 days learning trial for MWM. We have standardized this protocol and have published the findings using similar protocols (Yesudhas et al 2020) which is comparable to Guy B. Mulder et al 2003.
During the training session, each animal was systematically released from all directions clockwise (the releasing point was always new from the lasted releasing point N-E-S-W). Also, in the probe test, each animal was released from a common novel starting position which is the next point from the previous releasing point of learning.
1.4: What is the time between behavioral tasks? For instance, how many days after the MWM did you wait before doing the cued RAM?
Response: The time intervals between different behaviour experiments were 24 hours. The cued RAM was performed 1 day after MWM.
1.5: Line 193: should be micrometer, not micromolar (The capital M denotes Molarity, not meter as in distance)
Response: We thank reviewer 1 for indicating the mistake. We have rectified the errors as per the advice.
1.6: Line 269: AChE leads to the breakdown of acetylcholine into acetic acid and choline; can the authors please modify this or expand on what they currently have in place? Authors also state in line 518 that Ach breaks down into choline and acetate – so please rework these sections
Response: We thank reviewer 1 for the suggestion. The statements regarding the AChE and Ach have been revised.
1.7. Lines 287-295 could all be moved and incorporated into the methods section; the results should start discussing what was actually found
Rephrasing of your results to make it more clear: (e.g. Lines 299-300: As a result, aspirin-treated mice spent significantly more time in the novel object zone than the control group). Also, please add the p values for these results and appropriate t statistics as well (t value, degrees of freedom, etc.) You also state the units in the figure, but you should also state that these are seconds when you’re reporting the means (and I’m assuming you’re also reporting standard deviations? Or are those SEMs? Please clarify).
Response: We agree with the opinion of reviewer 1. We have moved Lines 287-295 from the result to the methodology. We rephrased the statements in the result. Previously, we added mean value ± standard deviations (SD) in the data. Now we have revised the data with mean value ± standard error mean SEM with units as per the suggestions. In addition, we have included the statistical test, significance, and p values in the figure legends.
1.8 Lines 313-319 could all be moved and incorporated into the methods section; the results should start discussing what was actually found. P values for all given analyses (latency, probe, reversal) are needed in your results; additionally, all F statistics should be provided as well (degrees of freedom, F value, etc.)
Response: We thank reviewer 1 for the suggestions, Lines 313-319 have been moved to the methodology we have added the mean ± SEM with p values and indicated the level of statistical significance which provides a complete and sufficient overview of the data regardless of the F statistics.
1.9 For figure 2, In figure captions, the authors should denote what the astericks (**, ***) mean with a legend, **p < 0.01, ***p < 0.001 (for instance).
Response: We thank reviewer 1 for the suggestions, we have added the P values in the figure legends.
1.10. Line 335: You state aspirin mice took 112 +/- 8; did you mean 12 or 11? This is a very large jump
Response: We thank reviewer 1 for pointing out the mistake. We express our sincere apology for the typographical error. It is 12+/- 8. We rectified the mistakes in the revised manuscript.
1.11. (Line 345 – it should be named 3.3 since this is following MWM which is 3.2). The researchers absolutely need to provide p values as well as t test results in reporting of results.
Response: We thank reviewer 1 for the suggestions. We have redefined the numbering of subheadings and we have added the p values obtained from the t-test in the results as well as figure legends.
1.12. For figure 3C, if there is no data to present for day 4, the authors should remove 4 from the x axis. In figure captions (this and the others), the authors should denote what the astericks (*) mean with a legend, *p < 0.05 (for instance).
Response: We thank reviewer 1 for the suggestion, we have removed day 4 from the x-axis in Fig 3 and we have added the P values in the results as well as figure legends
1.13 The authors should also be sure to put in the units of measurement in all reported results (for instance, 250 +/- 99 – what are the units being displayed here)?
Response: We thank reviewer 1 for the suggestions, we have added the units in the result part.
1.14. In results, the specific figure (for example, Fig 5a, Fib 5b) should be stated after discussion of that specific result, instead of just saying (Fig 5) at the very end of the result paragraph
Response: We thank reviewer 1 for the advice. We have added specific figure numbers in the respective statements in the revised manuscript.
1.15. Line 547: “…reveal the mechanisms responsible for the …”
Response: We thank reviewer 1 for the remark. We have corrected the mistake and rephrased the sentence as advised.
1.16. Overall comment: authors need to remark on 1) why female mice were not also included, and 2) has there been any research investigating aspirin given to female experimental mice or female human subjects?
Response: We thank reviewer 1 for the insightful clarification. Please refer to the response to question 1.1: As female mice exhibit varying phases of oestrus cycle, there could be chances for differential regulation of adult neurogenesis and neurocognitive function at the time of the experiment. Eventually, some reports indicate that aspirin therapy is less effective in women than in men. Hence this study has been restricted and devoted to assess the effect of aspirin in male mice. However, future studies are required to validate the effect of aspirin on neurodegenerative plasticity in female animals too. We incorporated a segment of text to cover this view in the discussion part.
In addition, we have revised the discussion and included a description of the possible effect of aspirin on female subjects. We updated the references sections too.
Reviewer 2 Report
Review for Manuscript- brainsci-2478587
The manuscript by Jemi Feiona Vergil Andrews et al., entitled " A mild dose of aspirin promotes hippocampal neurogenesis and working memory in experimental aging mice" is the research report finding that the effect of aspirin on spatial memory in correlation with the regulation of hippocampal neurogenesis and microglia in the brains of aging experimental mice. The authors reported that this study revealed that aspirin facilitates pro-neurogenic effects contributing to enhancing working memory. The manuscript is well-written and the cited references are appropriate. However, some remarks should be taken by authors under consideration before paper publication. The manuscript needs major revision before its final publication.
Comments:
1. Authors should include the schematic diagram for the experimental design. It will help the readers understand the experimental time points.
2. Authors should correct the sub-section numbers in the materials and methods section.
3. Authors should include the detailed procedure for the Aspirin preparation (eg., In what solution aspirin was dissolved and how much volume was prepared?)
4. Authors should explain how they choose 60 mg Aspirin as a low dose for the treatment.
5. Authors should include two more experimental groups in this study as Aspirin low and high doses.
6. In brain fixation overnight post-fixing with 4% PFA and 1 week in 30% sucrose is largely atypical. Was a preservative added to the sucrose solution?
7. Authors should confirm whether the brains were sectioned coronal or sagittal. The IHC images were looks like brains were sectioned in a coronal section.
8. In the statistics section, authors should use the standard error mean (SEM) for the statistical analysis.
9. In Figure 4, the authors wrongly mentioned the DAPI and DCX images. It should be corrected.
10. In Figure 5, the authors should include the Iba1 intensity measurement.
11. Authors should correct the sub-section numbers in the Results section.
Author Response
Reviewer 2
The manuscript by Jemi Feiona Vergil Andrews et al., entitled " A mild dose of aspirin promotes hippocampal neurogenesis and working memory in experimental aging mice" is the research report finding that the effect of aspirin on spatial memory in correlation with the regulation of hippocampal neurogenesis and microglia in the brains of aging experimental mice. The authors reported that this study revealed that aspirin facilitates pro-neurogenic effects contributing to enhancing working memory. The manuscript is well-written and the cited references are appropriate. However, some remarks should be taken by authors under consideration before the paper publication. The manuscript needs major revision before its final publication.
Response: We extend our sincere thanks to reviewer 2 for the overall positive statements and insightful remarks. We have addressed all the questions and implemented all suggestions.
- 1.Authors should include the schematic diagram for the experimental design. It will help the readers understand the experimental time points.
Response: We agree with the suggestion of reviewer 2 and we have included a schematic diagram for the experimental design.
2.2. Authors should correct the sub-section numbers in the materials and methods section.
Response: Thanks for the suggestion, we have corrected the sub-heading numbers in the revised manuscript
- 3.Authors should include the detailed procedure for the Aspirin preparation (eg., In what solution aspirin was dissolved and how much volume was prepared?)
Response: We agree with the opinion of the reviewer.
325 mg of dissolvable form of aspirin (Disprin tablet, Reckitt Benckiser Healthcare India PVT Ltd) was thoroughly mixed in 500 mL of lukewarm water. Each mouse with around 32-35 grams has been estimated to consume around 3 ml of aspirin solution daily which is equivalent to a daily dose of aspirin (around 60 milligrams (mg) /kilogram (kg) body weight (BW)). We updated the methodology section to cover the preparation and treatment of aspirin.
2.4. Authors should explain how they choose 60 mg Aspirin as a low dose for the treatment.
Response: Thanks for the meaningful comment by reviewer 2. Based on the literature study, we have fixed a possible dose of aspirin as per the discussion made in the animal ethical committee. Based on the previous reports, we have considered 60mg Aspirin /kg BW in this study (T. Wang 2010, Jeremy W. Smith et al 2002 and Soomaayeh Heysieattalab et al 2021, Antoine K. Kandeda et al 2021)
2.5. Authors should include two more experimental groups in this study as Aspirin low and high doses.
Response: We thank the reviewer for the suggestion. While a low dose might be less effective, a high dose of aspirin might be associated with adverse effects. Since we have considered 60mg of aspirin /kg BW as the optimum and effective dose in this candidate approach. We modestly believe that additional studies may not provide a major change to the outcome of the results in the present study and it requires additional ethical permission from the ethical committee and an unforeseen long time. Indeed, we have included a segment of text to compare and cover the possible effect of low and high doses of aspirin using available literature. Eventually, we have a plan to execute a future study using female mice, in which we will extend the study to cover and address all these aspects.
2.6. In brain fixation overnight post-fixing with 4% PFA and 1 week in 30% sucrose is largely atypical. Was a preservative added to the sucrose solution?
Response: We thank reviewer 2 for raising the clarification. The protocol for tissue fixation and cryopreservation have been well-established protocols and routinely practiced for the cryosections of the brain in our lab and other reputed labs (S Couillard-Despres 2009, L Torner, 2029, Marschallinger et al 2020, Kandasamy et al 2010 Kandasamy et al 2014., Selvaraj et al 2023). Overnight post-fixing with 4% PFA is necessary for the complete fixation of the whole brain. As sucrose is a non-cell permeable cryoprotectant, brains in 30% sucrose for 1 week at 4°C is an ideal and optimum time that is needed for the samples to be submerged. In case of a long-time preservation, sodium azide would be added as a preservative in sucrose solution however, it appears to interfere with the outcome of the immunohistochemical (IHC) procedure. For a short time, we used a sterile phosphate solution to prepare 30% sucrose. Then brains were immediately cryosectioned and cryopreserved. We have revised the methodology section.
2.7. Authors should confirm whether the brains were sectioned coronal or sagittal. The IHC images were looks like brains were sectioned in a coronal section.
Response: We thank reviewer 2 for the clarifications, we assure that the representative brain images in the manuscript were sagittal sections, while the DG of the hippocampus from the coronal section and sagittal sections would provide nearly a similar appearance (for example, please refer Molecular Biology Reports 49(3):3, 2022, DOI: 10.1007/s11033-022-07313-4). The detailed procedure of cryosectioning has been mentioned in the methodology part.
2.8. In the statistics section, authors should use the standard error mean (SEM) for the statistical analysis.
Response: We thank reviewer 2 for the comments and suggestion, the values were represented as mean ± standard deviation (SD). As per the suggestion, we have changed the data values to mean ± standard error mean (SEM).
2.9 In Figure 4, the authors wrongly mentioned the DAPI and DCX images. It should be corrected.
Response: We thank reviewer 2 for pointing out the mistake, and we apologize for the error. We have changed the label and have rectified the mistake in Figure 4.
2.10. In Figure 5, the authors should include the Iba1 intensity measurement.
Response: We thank reviewer 2 for the question we have added the methodology and results of Iba1 intensity in fig 5C.
2.11. Authors should correct the sub-section numbers in the Results section.
Response: We thank reviewer 2 for the remark, we have corrected the sub-section numbers in the manuscript.
Round 2
Reviewer 1 Report
Authors still have not provided appropriate T and F statistics to accompany the reported findings. While it's appreciated that significance values and p-values have been added, there needs to be T(df) = x and F(x1, x2) = y values present throughout the results to accompany the p values.
The discussion section lacks tie in with the behavioral results that the authors present. The discussion section does a good job at relating aspirins biochemical effects to reductions in inflammation and decreases in negative effects. However, it doesn't take a look at the behavioral effects of aspirin and tie it in with existing behavioral literature.
The readability of the manuscript is sufficient; however, I would ask that the authors provide the manuscript to someone that has a good grasp of the English language to read through it before final processing is done.
Author Response
Reviewer 1
1.1 Authors still have not provided appropriate T and F statistics to accompany the reported findings. While it's appreciated that significance values and p-values have been added, there needs to be T(df) = x and F(x1, x2) = y values present throughout the results to accompany the p values.
Response: We express our sincere thanks to reviewer 1 for the constructive remark. We felt that adding the degree of freedom and the F values in the manuscript is unusual/rare. However, we agree with reviewer 1 and we have added df and F values in the revised manuscript with the help of the subject experts.
1.2 The discussion section lacks tie in with the behavioral results that the authors present. The discussion section does a good job at relating aspirins biochemical effects to reductions in inflammation and decreases in negative effects. However, it doesn't take a look at the behavioral effects of aspirin and tie it in with existing behavioral literature.
Response: We admit the valid point of reviewer 1. We extended the discussion part to cover the behavioral aspect of aspirin in the revised manuscript and updated the reference section as per the suggestions.
Comments on the Quality of English Language
The readability of the manuscript is sufficient; however, I would ask that the authors provide the manuscript to someone that has a good grasp of the English language to read through it before final processing is done.
Response: We thank reviewer 1 for the appreciation and further suggestions. The manuscript has been checked by a language expert and the grammatical errors and typos have been rectified in the revised manuscript.
Reviewer 2 Report
The manuscript is well-revised and the cited references are appropriate. The manuscript is suitable for the final publication.
Author Response
Reviewer 2
2.1 The manuscript is well-revised and the cited references are appropriate. The manuscript is suitable for the final publication.
Response: We sincerely thank and appreciate reviewer 2 for constructive remarks, insightful suggestions, and valuable time which helped us to improve the content of the manuscript. We extend our sincere thanks to reviewer 2 for recommending the manuscript for final publication.